# The treatment outcomes of epilepsy and its root causes in children attending at the University of Gondar teaching hospital: A retrospective cohort study, 2018

Addisu Beyene[1], Agumas Fentahun Ayalew[2]*, Getasew Mulat[3], Ayele Simachew Kassa[4], Tigabu Birhan[5]

**1** Department of Pediatrics and Child Health, College of Medicine and Health Sciences, University of Gondar, Gondar, Ethiopia, **2** Family Guidance Association of Ethiopia, Bahir Dar Model Sexual and Reproductive Health Clinic, Bahir Dar, Ethiopia, **3** GAMBY Medical and Business College, Bahir Dar city, Ethiopia, **4** Department of Adult Nursing, Bahir Dar University, Bahir Dar, Ethiopia, **5** Bahir Dar Health Science College, Bahir Dar, Ethiopia

* fentahun143@gmail.com

## Abstract

### Background

Epilepsy is the most common chronic neurologic disorder which affects an estimated 10.5 million children worldwide. Despite the burden, the scarcity of study held in Ethiopia. Hence, the aim of this study was to assess the treatment outcomes of epilepsy and its root causes in children with epilepsy.

### Methods

A hospital-based retrospective cohort study was conducted from October 10/2017 up to October 10/2018. A total of 210 study participants who fulfilled the eligibility criteria were included in the study. A structured interviewer-administered questionnaire with a document review was used to collect data. The data were entered into Epi Info version 7.2.1 and analyzed using SPSS version 21. Descriptive statistics were computed. Simple logistic analysis was run (at 95% CI and p-value < 0.05) to identify factors associated with treatment outcome.

### Result

210 eligible patients with epilepsy were recruited for the study. About half of the respondents were females and the majority was within the age group of 5–10 years. Phenobarbital has been the most frequently prescribed drugs and thirteen percent of patients were in the escalation phase of treatment. Eight percent of the study participants had poor adherence to the treatment regimen. About six percent of the study subjects were suffering from an uncontrolled seizure.

**Data Availability Statement:** All relevant data are within the manuscript and its Supporting Information files.

**Funding:** The author(s) received no specific funding for this work.

**Competing interests:** The authors have declared that no competing interests exist.

**Abbreviations:** AED, Anti-Epileptic Drugs; AOR, Adjusted Odds Ratio; CI, Confidence Interval; COR, Crude Odd Ratio; UOG, University of Gondar.

Being a female child (AOR = 2. 21; 95%CI: 1.11, 4.41) and excellent adherence to anti-epilepsy treatment (AOR = 4. 51; 95%CI: 1.53, 13.42) were significantly associated with treatment outcome.

## Conclusion

This study revealed that many children were suffering from uncontrolled seizure and escalation therapy is being exercised. Being a female child and adherence to anti-epilepsy treatment were significantly associated with treatment outcome. Therefore, attention should be given to adherence counseling to convey a better treatment outcome.

## Introduction

Epilepsy is the most common chronic neurological disorder which affects an estimated 10.5 million children worldwide, and 80% of them live in developing countries, often accompanied by physical and cognitive disability[1–3]. Most children may have at least one epileptic seizure that is recommended for anti-epileptic drugs; which are prescribed according to episodes and severity of seizures [4]. Studies are highly important that initiate to develop methods towards epilepsy treatment and to evaluate the effective strategies for the therapeutic purpose [5,6]. Various studies revealed that having different treatment protocols, inaccurate epilepsy diagnosis, inappropriate health care providers, lack of modern diagnostic technology, delay in seeking health care, and lack of knowledge were the identified factors of treatment outcome of epilepsy in developing countries [7,8]. Despite epilepsy is the crucial public health problem [9], there are scanty studies on the treatment outcomes of epilepsy in children in Ethiopia in general, and in the study area in particular. Therefore, this study was designed to determine the treatment outcomes of epilepsy and its predictors in children attending at the University of Gondar teaching hospital.

## Methods

### Study design, period and area

A retrospective cohort study was conducted from October 10th, 2016 to October 10th, 2018 at the University of Gondar (UOG) hospital, which is 727 km away from Addis Ababa, the capital city of Ethiopia, in the North West direction. The hospital is a tertiary teaching hospital and one of the oldest institutes of Ethiopia serves for the whole of the North Gondar population. The hospital has a pediatrics neurologic follow up clinic which is delivering service for about 635 patients monthly, among which one-third of the cases are accounted for by patients with epilepsy. Neurologic patients were visiting the clinic once per week.

### Eligibility criteria

Clinically diagnosed Patients with epilepsy whose age is less than eighteen years and attending at the UOG hospital of pediatrics neurologic follow-up clinic were included. While patients with epilepsy who lost follow up of treatment before 3 months or new patients who commence the treatment within three months of the start of the survey were excluded.

## Sample size

All patients with epilepsy who fulfilled the eligibility criteria were included.

## Operational definitions

**Epilepsy**:Two or more unprovoked seizures occur in a time frame of >24 hours.

**Seizure**: Is a transient event of signs as well as indications coming about because of unusual over the top or synchronous neuronal action in the Brain.

**Well-controlled seizure**: Maximum of one seizure episode in the last 3 months after the start of treatment.

**Good control seizure**: Maximum of three seizure episodes in the last 3 months after the start of treatment.

**Poorly controlled seizure**: Maximum of nine, and a minimum of four seizure episodes in the last three months after the start of treatment.

**Uncontrolled seizure**: Ten and above seizure episodes in the last three months after the start of treatment.

**Treatment outcome**: Patients who had a maximum of three seizure episodes in the 3 months after the start of anti-epileptic treatment, it was considered as having a successful treatment outcome. However, if the patient had four and above seizure episodes in the 3 months after the start of anti-epileptic treatment, he/she had not successful treatment outcome.

**Excellent Adherence**: If the patient took more than 90% of his/her monthly medication.
**Good Adherence**: If the patient took more than 85% of his/her monthly medication.
**Poor Adherence**: If the patient took less than 85% of his/her monthly medication.

## Data collection tools and procedures

The study was undertaken in the outpatient department of pediatrics neurologic follow-up clinic. A structured questionnaire was designed and prepared in English and then translated into Amharic (local language). One day training was given for the enumerators by the principal investigator. Four data collectors, among this 1 resident, 1 general practitioner and 2 trained interns have participated. To collect data individual patient's medical cards were overlooked.

## Data processing and analysis

Data were checked for completeness, coded, entered and cleaned using Epi Info version 7.2.1 and exported to SPSS version 21.0 for further analysis. Descriptive statistics such as frequencies, percentages, mean and standard deviation were computed. Binary logistic regression was fitted to identify factors associated with treatment outcome and those variables with p-value < 0.25 were fitted to multivariate logistic regression analysis. Odds ratio with the corresponding 95% confidence interval and p-value less than 0.05 were considered statistically significant.

## Ethical considerations

Ethical clearance was obtained from the Institutional Ethical Review Board (IRB) of the University of Gondar. A permission letter was written to the hospital administrations from the university. The purpose of the study was explained to parents/caregivers and informed written consent was obtained from them all the purposes, notifying that they have the right to refuse or stop at any point of the interview. Confidentiality of information was maintained by avoiding possible any personal identifier from the questionnaire.

## Results

### Socio-demographic characteristics of patients

A total of 210 patients with epilepsy were included in the study. The demographic data revealed that the number of male and female patients was almost in equal proportion. The mean age of the study members was 8.89 (±4.3) years. The majority, (44.8%) of caregivers were fathers (**Table 1**).

### Treatment outcomes of patients with epilepsy

About one-fourth of the study subjects were commonly used dual therapy in the management of seizure. Regarding anti-epileptic drugs, 71% of the respondents have used phenobarbitone and 13.3% of patients were in the escalation phase of treatment.

Eight percent of the study participants had poor adherence to the treatment regimen. About six percent of the study subjects were suffering from an uncontrolled seizure. The mean duration of time to control seizure was 2.79 (±1.9) years. Forty-eight percent of patients reported that they had experienced adverse effects from their AED therapy. Of these, a behavioral abnormality was the commonest reported adverse effect (**Table 2**).

### Factors associated with treatment outcome of patients with epilepsy

Multivariate logistic regression output indicated that the sex of the child and adherence to treatment was significantly associated with treatment outcome.

The likelihood of developing a successful treatment outcome in females (AOR = 2.21; 95% CI: 1.11, 4.41) was 2.21 times higher than those within males.

The likelihood of developing a successful treatment outcome in patients with excellent adherence (AOR = 4. 51; 95%CI: 1.53, 13.42) was about 4.5 times higher than those with poor adherence. (**Table 3**).

**Table 1. Socio-demographic characteristics of patients with epilepsy at the University of Gondar hospital, pediatric neurologic follow-up clinic, Northwest Ethiopia, 2018 (n = 210).**

| Variable | Frequency (N) | Percentage (%) |
|---|---|---|
| **Sex of the child** | | |
| Male | 106 | 50.5 |
| Female | 104 | 49.5 |
| **Age of the child (in years)** | | |
| [1–5] | 53 | 25.2 |
| [5,10] | 82 | 39.0 |
| [10,15] | 62 | 29.5 |
| [15,18] | 13 | 6.2 |
| **Caregiver** | | |
| Father | 94 | 44.8 |
| Mother | 83 | 39.5 |
| Brother | 12 | 5.7 |
| Sister | 7 | 3.3 |
| Others[#] | 14 | 6.7 |

#-uncle, aunt, grandparent

**Table 2. Treatment outcomes of patients with epilepsy at the University of Gondar hospital, pediatric neurologic follow up clinic, Northwest Ethiopia, 2018.**

| Variable | Frequency (N) | Percentage (%) |
|---|---|---|
| **Mode of therapy used** | | |
| Mono-therapy | 154 | 73.3 |
| Dual therapy | 51 | 24.3 |
| Triple therapy | 5 | 2.4 |
| **Type of anti-epileptic drugs prescribed** | | |
| Phenobarbitone | 149 | 71.0 |
| Phenytoin | 90 | 42.9 |
| Valproic acid | 26 | 12.4 |
| Carbamazepine | 5 | 2.4 |
| **Phase of therapy** | | |
| Maintenance phase | 166 | 79 |
| Escalation phase | 28 | 13.3 |
| Tapering phase | 16 | 7.6 |
| **Seizure control status** | | |
| Good | 162 | 77.1 |
| Poor | 35 | 16.7 |
| Uncontrolled | 13 | 6.2 |
| **Adherence status to antiepileptic drugs** | | |
| Excellent adherence | 148 | 70.5 |
| Good | 46 | 21.9 |
| Poor | 16 | 7.6 |
| **Reported adverse effects of antiepileptic drugs** | | |
| Yes | 100 | 47.6 |
| No | 110 | 52.4 |
| **Adverse effects of antiepileptic drugs** | | |
| Behavioral abnormality | 24 | 23.8 |
| Gum hyperplasia | 21 | 21.4 |
| Skin rash | 6 | 6.2 |
| Ataxia | 3 | 3.3 |
| Drowsiness | 3 | 2.9 |
| Drug overdose | 2 | 2.4 |

## Discussion

Scientific investigations on the outcome of epileptic treatment and associated factors among children are necessary to design appropriate intervention strategies. This finding revealed that about 73% the participants responded to mono-therapy, which was greater than the study conducted in Scotland, 63.7% [10], in France, Germany, and the United States through the market research company InforMed Insight, UK, 50% [11] and Ayder comprehensive specialized hospital, 46.6%[12]. This might be due to this study conducted among a smaller sample size (n = 210) relatively than those above three studies. whereas it was lower than previous studies conducted in Gondar among adult epileptic outpatients [13]. The possible justification might be this study was conducted among child epileptic patients while the latter was conducted on adult epileptic outpatients. Regarding users of anti-epileptic drugs, phenobarbital was commonly prescribed than other anti-epileptic drugs.

**Table 3. Factors associated with treatment outcome in epileptic patients in the University of Gondar hospital, pediatric neurologic follow up clinic, in Northwest Ethiopia, 2010 (n = 210).**

| Variables | Treatment Outcome | | COR (95%CI) | AOR (95%CI) | P-value |
|---|---|---|---|---|---|
| | Successful N (%) | Not Successful N (%) | | | |
| **Educational status of the caregiver** | | | | | |
| No formal education[@] | 110 (79.7) | 28 (20.3) | 1.00 | 1.00 | 1.000 |
| Primary school | 30 (76.9) | 9 (23.1) | 0.85 (0.36,1.99) | 0.75 (0.28,1.96) | 0.552 |
| Secondary school | 11 (57.8) 11 (78.6) | 8 (42.2) 3 (21.4) | 0.35 (0.13,0.95) 0.93 (0.24,3.57) | 0.25 (0.78,0.81) 0.67 (0.15,2.93) | 0.210 0.590 |
| **Age of child (in years)** | | | | | |
| <5[@] | 36 (67.9) | 17 (32.1) | 1.00 | 1.00 | 1.000 |
| 5–10 | 66 (80.5) | 16 (19.5) | 1.95 (0.88,4.31) | 1.50 (0.61,3.71) | 0.376 |
| 11–15 | 51 (82.3) | 11 (17.8) | 2.19 (0.92,5.23) | 2.14 (0.82,5.62) | 0.122 |
| >15 | 9 (69.2) | 4 (30.8) | 1.10 (0.29,3.94) | 1.31 (0.24,7.08) | 0.751 |
| **Sex of child** | | | | | |
| Male[@] | 75 (70.8) | 31 (29.2) | 1.00 | 1.00 | 1.000 |
| Female | 87 (83.7) | 17 (26.3) | 2.12 (1.10,4.12) | 2.21 (1.11,4.41) | 0.025 |
| **Adherence to anti—epileptic drugs** | | | | | |
| Excellent | 122 (82.4) | 26 (17.6) | 4.69 (1.61,13.65) | 4.51 (1.53,13.42) | 0.006 |
| Good | 32 (69.6) | 14 (29.4) | 2.29 (0.71,7.32) | 1.98 (0.60,6.52) | 0.259 |
| Poor[@] | 8 (50.0) | 8 (50.0) | 1.00 | 1.00 | 1.00 |

[@]reference category

Regarding adherence to treatment, about 8% of the study participants had poor adherence. Whereas about 6% of the respondents were suffering from an uncontrolled seizure.

Adherence to anti-epileptic treatment was one of the identified significant predictors associated with treatment outcome. The likelihood of having successful treatment outcomes in patients with excellent adherence was about 4.5 times higher than those with poor adherence to treatment. This is consistent with one study from Ayder comprehensive, specialized hospital [12] and Nigeria [14].

Gender was also the other factor associated with anti-epileptic treatment outcome. Females were about 2.21 times more likely to have successful anti-epileptic treatment outcomes than males. This finding is supported by another study[15]. This could be explained by the fact that females give better due attention to every aspect of their life, and they might think everything critically and they might also afraid of the complications of missing the anti-epilepsy medications, which in fact might be associated with better success of treatment outcome of epilepsy.

## Limitation of the study

Since the study was retrospective study there was a difficulty of obtaining full information from because of the difficulty of recall for those who had a longer duration of follow up.

Use of WHO operational definition of treatment outcome and the difficulty of comparing it with other settings to see the patient's response.

Lack of imaging modalities like EEG and other Neuro-imaging to reach to a specific type of seizure and etiology.

Another challenge was frequent switching of patients' drug because of running out of medications what the patient was getting and the difficulty of concluding the patient was adherent to medication.

## Conclusion and recommendation

The study also showed that more than 3/4[th] of patients have controlled seizure with 2.9 ± 1.9 years of the treatment period. Child sex and adherence to anti-epilepsy treatment were significantly associated with the success of treatment outcome. Caregivers should be continuously counseled on proper treatment adherence to improve the treatment outcome of children and male patients should also get attention regarding their treatment outcome.

## Supporting information

**S1 Data.**
(SAV)

## Acknowledgments

The authors extend their acknowledgment to the University of Gondar department of pediatrics and child health for their knowledge and experience sharing; enumerators for their extensive data collection; and our study participants for their time and patience.

## Author Contributions

**Conceptualization:** Ayele Simachew Kassa.

**Data curation:** Addisu Beyene.

**Formal analysis:** Addisu Beyene, Agumas Fentahun Ayalew, Getasew Mulat, Ayele Simachew Kassa, Tigabu Birhan.

**Methodology:** Addisu Beyene, Agumas Fentahun Ayalew, Getasew Mulat, Ayele Simachew Kassa, Tigabu Birhan.

**Software:** Agumas Fentahun Ayalew, Getasew Mulat, Tigabu Birhan.

**Writing – original draft:** Addisu Beyene.

**Writing – review & editing:** Agumas Fentahun Ayalew.

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
