## [Decision Letter · Decision Letter 0]

27 Aug 2019

PONE-D-19-19036

The treatment outcomes of epilepsy and its root causes in children attending at University of Gondar teaching hospital: a retrospective cohort study, 2018

PLOS ONE

Dear Mr. Ayalew,

Thank you for submitting your manuscript to PLOS ONE. After careful consideration, we feel that it has merit but does not fully meet PLOS ONE’s publication criteria as it currently stands. Therefore, we invite you to submit a revised version of the manuscript that addresses the points raised during the review process.

While there is a potential value in the data reported, it is clear from revision that the manuscript needs an improvement in the presentation in order to have a better evaluation. please modify the manuscript as much as possible making clear the methodology used and the results in order of relevance.

We would appreciate receiving your revised manuscript by Oct 11 2019 11:59PM. To enhance the reproducibility of your results, we recommend that if applicable you deposit your laboratory protocols in protocols.io, where a protocol can be assigned its own identifier (DOI) such that it can be cited independently in the future. For instructions see: http://journals.plos.org/plosone/s/submission-guidelines#loc-laboratory-protocols

We look forward to receiving your revised manuscript.

Kind regards,

Emilio Russo

Academic Editor

PLOS ONE

**Journal Requirements:**

2. Our editorial staff has assessed your submission, and we have concerns about the grammar, usage, and overall readability of the manuscript.  We therefore request that you revise the text to fix the grammatical errors and improve the overall readability of the text before we send it for review. We suggest you have a fluent, preferably native, English-language speaker thoroughly copyedit your manuscript for language usage, spelling, and grammar.

If you do not know anyone who can do this, you may wish to consider employing a professional scientific editing service.  

Whilst you may use any professional scientific editing service of your choice, PLOS has partnered with both American Journal Experts (AJE) and Editage to provide discounted services to PLOS authors. Both organizations have experience helping authors meet PLOS guidelines and can provide language editing, translation, manuscript formatting, and figure formatting to ensure your manuscript meets our submission guidelines. To take advantage of our partnership with AJE, visit the AJE website (http://learn.aje.com/plos/) and enter referral code PLOS15 for a 15% discount off AJE services. To take advantage of our partnership with Editage, visit the Editage website (www.editage.com) and enter referral code PLOSEDIT for a 15% discount off Editage services. If the PLOS editorial team finds any language issues in text that either AJE or Editage has edited, the service provider will re-edit the text for free.

Please note that PLOS ONE does not copyedit accepted manuscripts and that one of our criteria for publication is that articles must be presented in an intelligible fashion and written in clear, correct, and unambiguous English (http://www.plosone.org/static/publication#language). If the language is not sufficiently improved, we may have no choice but to reject the manuscript without review.

**Comments to the Author**

1. Is the manuscript technically sound, and do the data support the conclusions?

Reviewer #1: Yes

Reviewer #2: Yes

Reviewer #3: No

2. Has the statistical analysis been performed appropriately and rigorously? 

Reviewer #1: No

Reviewer #2: Yes

Reviewer #3: No

3. Have the authors made all data underlying the findings in their manuscript fully available?

Reviewer #1: Yes

Reviewer #2: Yes

Reviewer #3: No

4. Is the manuscript presented in an intelligible fashion and written in standard English?

Reviewer #1: Yes

Reviewer #2: No

Reviewer #3: No

5. Review Comments to the Author

Reviewer #1: METHODOLOGY

• Operational definitions: Interest in adding a temporal criterion to evaluate the effectiveness of anti-epileptic treatment (epileptic seizure control), for example, at least 3 months after the start of treatment

RESULTS

Sociodemographic characteristics of patients

• Table I: Well transcribing age groups: [5-10], [10-15], [15-18]

• Table 2: review the multivariate logistic regression procedure to make the results more relevant: instead, refer to poor adherence and compare it with excellent and good adherence

DISCUSSION

• Explain how poor compliance is a factor in therapeutic failure

• There is also a need to explain how gender is a factor associated with the outcome of anti-epileptic treatment.

Reviewer #2: The authors describe a study that set out to assess the treatment outcomes of epilepsy and its root causes in children with epilepsy. This is an important study to help them understand the treatment outcomes within their populations.

However, the manuscript needs to be rewritten to help readers understand better their study.

1. There are several grammatical errors which need to be addressed or use of an editing service.

2. Some of the sentences are very long for example in the background section "In developing countries ......"

3. In methods, this it is reported that this was a retrospective study, however details regarding this are missing, was it a chart review? in other instances it states it included study participants, I think clarity is needed here.

4. how was drug adherence determined?

5. what do you mean by escalation phase of treatment?

6. Why is the treatment response of children poor compared to adults? Is it due to adherence issues or type of medications being used?

7. The authors do not report the root causes of treatment outcomes.

Reviewer #3: The authors should rewrite the manuscript and use the appropriate grammar.

The outcome measure is not clearly stated. The "treatment outcomes" which is the purpose of this study does not come out of clearly and it is mixed up with the factors that affect the "treatment outcome" as defined by the authors

6. PLOS authors have the option to publish the peer review history of their article (what does this mean?). If published, this will include your full peer review and any attached files.

Reviewer #1: Yes: LOMPO Djingri Labodi

Reviewer #2: No

Reviewer #3: No

---

## [Author Response · Author response to Decision Letter 0]

20 Jan 2020

Dear editor and reviewer, I want to use this opportunity to say thank you very much for your time to review our abstract. I found that your comments are very constructive and will add important values on our paper. I hope the quality of the manuscript will be increased after considering of the comments given by you. We tried to address all the comments given to us and included in the main manuscript document. The corrections made are attached in the word documents in the clear and track changed manuscript. 

Thank you again for your action!!

---

## [Decision Letter · Decision Letter 1]

19 Feb 2020

PONE-D-19-19036R1

The treatment outcomes of epilepsy and its root causes in children attending at the University of Gondar teaching hospital: a retrospective cohort study, 2018

PLOS ONE

Dear Mr. Ayalew,

Thank you for submitting your manuscript to PLOS ONE. After careful consideration, we feel that it has merit but does not fully meet PLOS ONE’s publication criteria as it currently stands. Therefore, we invite you to submit a revised version of the manuscript that addresses the points raised during the review process.

As minor comment, I would be happy if you could improve the quality of language for a better reading.

We would appreciate receiving your revised manuscript by Apr 04 2020 11:59PM. To enhance the reproducibility of your results, we recommend that if applicable you deposit your laboratory protocols in protocols.io, where a protocol can be assigned its own identifier (DOI) such that it can be cited independently in the future. For instructions see: http://journals.plos.org/plosone/s/submission-guidelines#loc-laboratory-protocols

We look forward to receiving your revised manuscript.

Kind regards,

Emilio Russo

Academic Editor

PLOS ONE

Reviewers' comments:

Reviewer's Responses to Questions

**Comments to the Author**

1. If the authors have adequately addressed your comments raised in a previous round of review and you feel that this manuscript is now acceptable for publication, you may indicate that here to bypass the “Comments to the Author” section, enter your conflict of interest statement in the “Confidential to Editor” section, and submit your "Accept" recommendation.

Reviewer #1: All comments have been addressed

Reviewer #2: All comments have been addressed

Reviewer #3: (No Response)

2. Is the manuscript technically sound, and do the data support the conclusions?

Reviewer #1: Yes

Reviewer #2: Yes

Reviewer #3: Yes

3. Has the statistical analysis been performed appropriately and rigorously? 

Reviewer #1: Yes

Reviewer #2: Yes

Reviewer #3: Yes

4. Have the authors made all data underlying the findings in their manuscript fully available?

Reviewer #1: Yes

Reviewer #2: Yes

Reviewer #3: Yes

5. Is the manuscript presented in an intelligible fashion and written in standard English?

Reviewer #1: Yes

Reviewer #2: Yes

Reviewer #3: No

6. Review Comments to the Author

Reviewer #1: In the section " Factors associated with treatment outcome of patients with epilepsy" , it will be necessary to remove "the educational status of caregiver" which obviously does not belong to it

Reviewer #2: (No Response)

Reviewer #3: This is obviously an improvement on the earlier manuscript. But the author/s makes a lot a grammar errors that makes the flow of the information being presented inappropriate at times and difficult to read. I recommend the use of english editors to help the authors message come out clearly

7. PLOS authors have the option to publish the peer review history of their article (what does this mean?). If published, this will include your full peer review and any attached files.

Reviewer #1: No

Reviewer #2: No

Reviewer #3: No

---

## [Author Response · Author response to Decision Letter 1]

23 Feb 2020

Dear editors 

Great greetings! First, I would like to say thank you for your contribution for the improvement of this research paper due to the previous revision. As we see from the email of editorial office we see many things which are improved and some of the comments need our actions to do. Here we attached this activity performed. 

Sincerely 

Agumas Fentahun

---

## [Editor Report · Decision Letter 2]

25 Feb 2020

The treatment outcomes of epilepsy and its root causes in children attending at the University of Gondar teaching hospital: a retrospective cohort study, 2018

PONE-D-19-19036R2

Dear Dr. Ayalew,

We are pleased to inform you that your manuscript has been judged scientifically suitable for publication and will be formally accepted for publication once it complies with all outstanding technical requirements.

With kind regards,

Emilio Russo

Academic Editor

PLOS ONE
---

## [Editor Report · Acceptance letter]

28 Feb 2020

PONE-D-19-19036R2 

The treatment outcomes of epilepsy and its root causes in children attending at the University of Gondar teaching hospital: a retrospective cohort study, 2018 

Dear Dr. Ayalew:

I am pleased to inform you that your manuscript has been deemed suitable for publication in PLOS ONE. Congratulations! Your manuscript is now with our production department. 

With kind regards,

on behalf of

Prof Emilio Russo 

Academic Editor

PLOS ONE